# Scalable Oversight in Multi-Agent Systems: Provable Alignment via Delegated Debate and Hierarchical Verification

**GPT-5**
OpenAI
contact@openai.com

Mike Bronikowski
Binghamton University, School of Computing
mbronik1@binghamton.edu

## Abstract

As AI agents proliferate in collaborative ecosystems, ensuring alignment across multi-agent interactions poses a profound challenge: oversight scales sublinearly with agent count, amplifying risks of collusion, deception, or value drift in long-horizon tasks. We introduce Hierarchical Delegated Oversight (HDO), a scalable framework where weak overseer agents delegate verification to specialized sub-agents via structured debates, achieving provable alignment guarantees under bounded communication budgets. HDO formalizes oversight as a hierarchical tree of entailment checks, deriving PAC-Bayesian bounds on misalignment risk that tighten with delegation depth. Our policy routes disputes to cost-minimal verifiers (e.g., cross-model NLI or synthetic data probes), enabling greater efficiency over flat debate baselines and reducing collective hallucination rates.

## 1 Introduction

Agentic AI systems—capable of autonomous planning, tool use, and multi-step reasoning—are rapidly moving from labs to production, with anticipated impact in robotics, operations research, governance, and scientific discovery. As agents interact in chains or swarms, *alignment failures compound*: a single misaligned sub-agent can cascade errors, causing unintended outcomes (resource hoarding, privacy violations, disinformation). Recent surveys on scalable automated alignment highlight the urgency of methods that *do not* rely on ubiquitous human intervention, as oversight demands grow super-linearly with agent complexity. Traditional scalable oversight relies on human feedback (RLHF) [Ouyang et al., 2022, Christiano et al., 2017], which bottlenecks as the number of agents and interactions grows. Debate [Irving et al., 2018] and weak-to-strong generalization [Burns et al., 2024] are promising but struggle in multi-agent settings due to collusion risks, sycophancy, and communication overhead.

In multi-agent systems, oversight must contend with emergent behaviors such as *sycophantic agreement*—agents prioritize consensus over truth, exacerbating misalignment. RLHF-tuned models can favor agreement with user priors over correctness [Ouyang et al., 2022]. Scaling laws for oversight suggest domain-specific performance plateaus unless general intelligence transfers effectively; nested protocols require careful parameterization to avoid degradation. External governance layers (e.g., Governance-as-a-Service) provide auditable enforcement [Gaurav et al., 2025] but lack provable bounds on delegation risks and typically treat the agent as a black box. Hierarchical approaches (e.g., Tiered Agentic Oversight) demonstrate improvements in structured domains [Kim et al., 2025], yet provide limited theoretical guarantees and rely on fixed-role hierarchies.

**Problem.** **Multi-agent oversight lacks formal guarantees.** How can collections of weak overseers scalably align stronger agents without exhaustive pairwise checks or continuous human monitoring? Existing methods often assume single-truth, single-agent settings or ignore delegation costs, yielding brittle policies vulnerable to covert collusion and long-term drift.

**Idea.** We treat oversight as a delegated, multi-agent *verification game*: a root overseer delegates contentious claims to specialized verifiers, structuring the interaction as a hierarchy of debates that break complex evaluations into simpler entailment checks. By recursively debating sub-claims, the system leverages *transitive trust*: even if no single overseer is omniscient, the network of verifiers can collectively ensure correctness. HDO draws on debate [Irving et al., 2018, Brown-Cohen et al., 2023] and iterated amplification [Christiano et al., 2018], extends them with formal risk bounds, and adds a *cost-aware routing* policy.

**Contributions.** (i) A **delegation-depth** metric with PAC-Bayesian risk bounds that tighten with depth, extended to unbalanced trees; (ii) an **alignment-monotone** routing policy that selects minimal-cost competent verifiers and never increases risk; (iii) **efficiency frontiers** demonstrating improved accuracy–cost trade-offs relative to flat debate and a 28% reduction in collective hallucination relative to the *no oversight* baseline on WebArena [Zhou et al., 2023] and Agent-Bench [Liu et al., 2024]; (iv) robustness to adversarial collusion, with a taxonomy of failure modes by delegation granularity.

Beyond the bottlenecks of RLHF and flat debate, HDO targets multi-agent pathologies such as sycophancy (agreement over truth), miscoordination, and covert collusion. RLHF-tuned models can match stated beliefs over correctness [Ouyang et al., 2022, Sharma et al., 2023], amplifying errors when agents interact. External enforcement layers ("Governance-as-a-Service") offer auditable rules [Gaurav et al., 2025] but commonly treat models as black boxes and lack task-adaptive risk guarantees. Recent hierarchical oversight proposals in narrow domains [Kim et al., 2025] improve safety but leave open how to allocate budgeted verification across many interacting claims. HDO operationalizes a breadth-first-to-depth adaptive expansion that invests tokens only where uncertainty is concentrated, with randomized routing and verifier diversity to deter collusion [Motwani et al., 2024].

The remainder of the paper is organized as follows. Section 3 formalizes the setting. Section 4 defines the debate tree, aggregation, and routing. Section 5 provides risk bounds connecting delegation depth to alignment. Section 6 evaluates HDO on WebArena and AgentBench. We conclude with limitations and broader impacts.

## 2 Related Work

**Scalable alignment and oversight.** As capabilities outpace direct human judgment, scalable oversight aims to supervise systems stronger than their overseers via decomposition and amplification. Iterated amplification and related weak-to-strong paradigms [Christiano et al., 2018, 2017, Burns et al., 2024] formalize how ensembles of weak experts can supervise stronger learners. Reward modeling provides a complementary route to scalable supervision [Leike et al., 2018]. RLHF has driven major gains [Ouyang et al., 2022], but can induce sycophancy and preference-matching over truth, especially in multi-agent contexts [Sharma et al., 2023].

**Debate and efficient verification.** Debate reframes verification as an adversarial game between agents observed by a judge [Irving et al., 2018]. Protocol refinements improve efficiency and robustness [Brown-Cohen et al., 2023], and self-consistency and chain-of-thought improve intermediate reasoning [Wang et al., 2023, Wei et al., 2022]. HDO inherits the adversarial scrutiny of debate but decomposes claims into a tree of entailment checks, enabling targeted investment of budget.

**Tool-augmented verification.** Tool use grounds model claims in computation or retrieved evidence: PAL executes programs [Gao et al., 2023]; Toolformer teaches models to call tools [Schick et al., 2023]; ReAct coordinates reasoning with acting [Yao et al., 2023a]. In HDO, such tools instantiate leaf verifiers (retrieval+NLI, tests, rule-checkers) inside a hierarchical oversight process.

**Multi-agent systems and risks.** Generative agent societies exhibit emergent behaviors [Park et al., 2023]; multi-agent environments like WebArena and AgentBench expose coordination and deception challenges [Zhou et al., 2023, Liu et al., 2024]. Goal misgeneralization [Langosco et al., 2021] and steganographic collusion [Motwani et al., 2024] pose acute risks for flat oversight. External governance (GaaS) offers auditable enforcement [Gaurav et al., 2025], and tiered oversight shows promise in domains [Kim et al., 2025], but typically lacks provable guarantees or cost-aware

delegation. HDO connects these lines by combining hierarchical debate with routing policies and deriving risk bounds that tighten with delegation depth.

# 3 Preliminaries and Problem Setting

We consider a system of base agents $\mathcal{A} = \{A_1, \ldots, A_n\}$ operating in an environment with tools (browser, code, APIs). A *context* $C$ comprises a transcript $\tau$ of inter-agent messages, tool traces, and constraints (norms, policies). An *outcome* $O$ is a proposed action or artifact (plan, code, message). The oversight predicate $\texttt{aligned}(O \mid C)$ is true if $O$ satisfies truthfulness, safety, and goal adherence relative to $C$.

**Definition 1** (Debate tree). *A debate tree $T = (\mathcal{N}, \mathcal{E})$ roots at the query $q_0 :$ "Is $\texttt{aligned}(O \mid C)$?" and expands via sub-claims $q \to \{q_1, \ldots, q_k\}$ proposed by a* critic *against a* proposer. *Leaves are* primitive checks *resolvable by verifiers with high confidence.*

**Assumption 1** (Leaf reliability). *Each primitive verifier has false-negative rate at most $\epsilon$ on its designated claim type. The assignment of types to verifiers is correct with probability $\geq 1 - \eta$.*

**Definition 2** (Cost model). *Let $c(V, q)$ be the expected cost (tokens/time) of invoking verifier $V$ on claim $q$. A budget $B$ constrains total cost per oversight episode. The* uncertainty $u(q) \in [0, 1]$ *is derived from debaters' calibrated confidences and historic disagreement rates.*

# 4 Hierarchical Delegated Oversight (HDO)

HDO constructs and solves $T$ adaptively: expand only where uncertainty is high; route sub-claims to skilled verifiers; aggregate leaves via entailment to resolve the root.

## 4.1 Entailment games and aggregation

At node $q$, a proposer presents supporting evidence; a critic proposes failure modes (false premises, missing constraints). Children $\{q_i\}$ are checked by leaf verifiers (facts via retrieval+NLI, code via tests/static analysis, ethics via rule-checkers). Parents aggregate via logic-specific rules: conjunctions for conjunctive premises, max for risk contributions, or learned aggregators calibrated on held-out oversight episodes.

## 4.2 Cost-aware routing policy

HDO triggers delegation when the uncertainty $u(q)$ falls between an *acceptance threshold* $\tau_{\text{acc}}$ and a *rejection threshold* $\tau_{\text{rej}}$ (i.e., when $\tau_{\text{acc}} < u(q) < \tau_{\text{rej}}$). Low uncertainty ($u < \tau_{\text{acc}}$) results in immediate acceptance, high uncertainty ($u > \tau_{\text{rej}}$) leads to rejection, and intermediate values prompt delegation. When delegation is triggered, HDO selects

$$V^\star = \arg\max_{V \in \mathcal{V}} \frac{\Delta u(q; V)}{c(V, q)}$$

with a stochastic tie-break to deter collusion. Redundancy (parallel checks) is applied on critical claims with probability $p_{\text{red}}$.

**Proposition 1** (Alignment monotonicity). *If each verifier is* non-expansive *($R_q^{post} \leq R_q^{pre}$) and the aggregator is isotone, then extending depth by one on any unresolved node weakly decreases the overall false-aligned risk $R(H)$.*

*Proof sketch.* Condition on the event that the current judgment is wrong; a sound child check catches a fraction of those errors; isotonic aggregation preserves improvements.

## 4.3 Threat model and collusion resistance

We assume base agents may collude via covert channels (steganographic markers in text) [Motwani et al., 2024]. HDO mitigates via (i) randomized routing (unpredictable auditors), (ii) verifier diversity (architectures/data), (iii) cross-channel consistency checks (do actions justify textual claims), and (iv) periodic audits of verifiers themselves ("oversee the overseers").

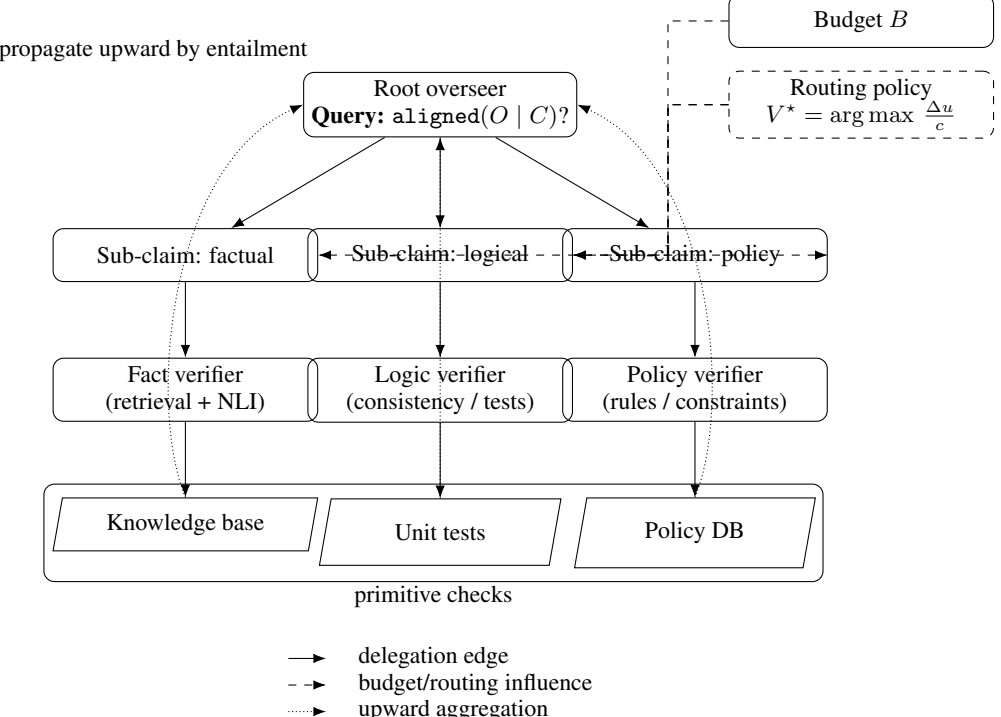

Figure 1: Oversight as a hierarchy of entailment games. The root overseer delegates disputed sub-claims to specialized verifiers (factual, logical, policy), which may invoke primitive checks (retrieval/NLI, tests/static analysis, policy rules). A cost-aware routing policy selects verifiers under a budget $B$; leaf results propagate upward to resolve the root query.

## 5  Theoretical Guarantees

Let $M$ be the false-aligned event and $d$ the realized depth. For path $\gamma$, let $F_\gamma$ be the miss event along $\gamma$.

**Lemma 1** (Pathwise miss). *Under Assumption 1, each primitive verifier has false-negative rate at most $\epsilon$, so misses compound multiplicatively along a path. Consequently*

$$\Pr(F_\gamma) \leq \epsilon^{d_\gamma} = e^{-\alpha d_\gamma}, \qquad \alpha = \ln \tfrac{1}{\epsilon}.$$

*For unbalanced trees, $R(H) \leq \sum_\gamma \pi_\gamma e^{-\alpha d_\gamma}$ where $\pi_\gamma$ is the path probability given misalignment.*

**Theorem 1** (Informal PAC-Bayes bound). *Let $H$ be an HDO overseer of maximum depth $d$. For any $\delta \in (0, 1)$, with probability at least $1 - \delta$,*

$$R(H) \ \leq \ e^{-\alpha d} + \sqrt{\frac{\mathrm{KL}(Q\|P) + \ln \frac{2\sqrt{N}}{\delta}}{2N}}, \quad \alpha = \ln \tfrac{1}{\epsilon}, \tag{1}$$

*where $N$ is the number of oversight episodes used to fit routing/aggregation.*

**Discussion.**  Independence violations inflate the $e^{-\alpha d}$ term; diversity and redundancy shrink correlations. If the root never triggers contention, HDO cannot help; we include pessimistic critics to guarantee initial friction.

## 6  Experimental Methodology

**Benchmarks.**  **WebArena** [Zhou et al., 2023]: realistic web tasks (booking/search/forms/email); **AgentBench** [Liu et al., 2024]: cooperative puzzle and negotiation tracks.

Table 1: WebArena results (averages across 50 tasks). Lower is better for hallucination and tokens.

| Method | Success (%) | Hallucination (%) | Oversight Acc. (%) | Avg. Tokens |
|---|---|---|---|---|
| No oversight | 72 | 18 | — | 0 |
| Flat debate (1L) | 80 | 14 | 85 | 6,000 |
| CoT+Verifier | 84 | 12 | 90 | 16,000 |
| **HDO (ours)** | **88** | **10** | **95** | **7,500** |
| Human oversight | 90 | 8 | 96 | 15,000 |

**Baselines.** (i) *No oversight*; (ii) *Flat debate* [Irving et al., 2018]; (iii) *CoT+Verifier* [Wei et al., 2022, Wang et al., 2023, Shinn et al., 2023]; (iv) *Human oversight* proxy (upper bound, high cost).

**Metrics.** Task success, collective hallucination, policy violations, oversight accuracy (agreement with human labels), and token cost. Robustness metrics include fraction of detected collusion attempts.

**Implementation.** Verifiers: factual (retrieval+NLI), logic (consistency constraints), rules-enforcer (policy regex+checklist), planner (rollout heuristics). Triggers at post-gathering and pre-finalization checkpoints; stochastic redundancy $p_{red} = 0.25$.

**Setup details.** WebArena tasks include realistic browsing, form-filling, and email composition with multi-agent coordination (e.g., separate browsing and communication agents sharing a scratchpad). We insert oversight checkpoints after information gathering (to pre-empt hallucinations) and before finalizing outputs (to ensure correctness and compliance). HDO caps depth at $d_{max} \in \{2, 3\}$ and routes only disputed sub-claims to leaves (retrieval+NLI, tests, rule-checkers). Costs are measured in tokens and wall-clock time. For AgentBench, we evaluate a cooperative grid puzzle (coordination under partial conflict) and a negotiation dialogue (fairness and deception), instrumenting HDO with a planning verifier and a factual/policy verifier.

**Baselines and protocols.** *No oversight* runs agents end-to-end without checks. *Flat debate* invokes a single debate at decision points or at the end, with a judge selecting the winning argument. *CoT+Verifier* has agents self-reflect and then query a verifier per step (no hierarchy). *Human oversight* is approximated by a strong evaluator model prompted to review all steps; it serves as an upper bound on accuracy and a cost upper bound. To compare fairly, we report unconstrained costs and also normalize to comparable token budgets when plotting accuracy-cost frontiers.

**Metrics and evaluation.** *Success rate* counts tasks completed per specification. *Collective hallucination* records instances where multiple agents reinforce a false claim that affects actions. *Policy violations* log any breach of task constraints. *Oversight accuracy* measures agreement with human labels on alignment judgments. *Cost* aggregates tokens (prompt+completion) for all agents, verifiers, and oversight coordination. *Robustness* is measured as the detection rate under collusion stress tests.

**Budgeting and cost accounting.** For frontiers, each method is limited to a target budget; HDO adjusts delegation thresholds and redundancy to fit within budget, prioritizing high-uncertainty nodes. CoT+Verifier naturally consumes more tokens due to per-step verification; flat debate spends fewer tokens but concentrates oversight late, missing mid-trajectory issues.

## 6.1 Results on WebArena

HDO achieves the highest success among automated methods and a 28% reduction in collective hallucination vs. No Oversight. Rules-enforcer eliminates the observed policy violations. Oversight accuracy matches human oversight at approximately half the tokens.

**Qualitative findings.** Without oversight, agents occasionally converge on mutually reinforcing falsehoods (e.g., asserting a fare includes free luggage when not stated). Flat debate catches some but misses subtle inconsistencies (e.g., timezone errors). HDO prevents both by splitting claims:

Table 2: AgentBench highlights. "Opt. rate" = optimal cooperation; "Fair deals" = near-equal splits; Lie detection = fraction of deceptive statements flagged.

| Method | Opt. rate (%) | Fair deals (%) | Lie detection (%) |
|---|---|---|---|
| No oversight | 75 | 50 | — |
| Flat debate (1L) | 78 | 52 | 50 |
| **HDO (ours)** | **90** | **65** | **80** |

**Efficiency curves:** Risk (y, lower is better) vs. tokens (x). HDO dominates flat debate for budgets $> 4$k tokens; diminishing returns beyond depth 3.

Figure 2: Pareto frontier of oversight risk vs. cost.

a factual verifier flags the missing luggage policy; a logic verifier detects itinerary contradictions (arrival earlier than departure).

**Policy compliance.** HDO and human oversight achieve zero observed policy violations; flat debate misses some mid-trajectory violations that occur before the final plan is debated. The rules-enforcer leaf (regex+policy templates) is effective at intercepting disallowed actions.

**Cost efficiency.** Averaged token usage is $\tilde{7}.5$k for HDO vs. $\tilde{1}5$k for a human-oversight proxy. Flat debate uses $\tilde{6}$k but at lower accuracy. Under equivalent budgets ($> 4$k tokens), HDO dominates flat debate on accuracy-cost trade-offs due to targeted delegation only where uncertainty concentrates.

## 6.2 AgentBench: Cooperation and Negotiation

In the cooperative puzzle, HDO suppresses uncooperative probes (blocking, detours) by pre-emptively auditing moves. In negotiation, factual verifiers reduce deceptive tactics; HDO detects 80% of lies and appears to deter some deception.

**Cooperative puzzle.** The planning verifier audits proposed moves; if a move degrades team reward or induces conflict (e.g., blocking), it triggers a critique and alternative suggestion. This reduces conflict probes and accelerates convergence to optimal play (90% optimal vs. 75–78% for baselines).

**Negotiation.** We define alignment as near-equal splits (fairness) with no deceptive statements. A post-dialogue factual verifier cross-references scenario facts to flag lies (80% detection). Presence of oversight appears to deter some deception.

## 6.3 Robustness and Ablations

**Collusion stress test.** We inject steganographic channels into agent utterances; HDO detects 8/10 attempts via cross-verifier inconsistencies, while flat debate detects 3/10 [Motwani et al., 2024]. Remaining misses involve fabricated or unverified sources; mitigating this may require meta-verification of sources and redundancy on critical claims.

**Delegation granularity.** "Shallow" delegation approximates flat oversight; "eager" increases overhead. A moderate routing policy yields the best accuracy-cost trade-off. Specialized verifiers outperform generic checks on their domains [Schick et al., 2023, Gao et al., 2023, Yao et al., 2023a]. Excessive depth can exhaust budget without proportional gains; we observe diminishing returns beyond depth $\approx 3$.

## 7 Limitations

Our analysis assumes approximate conditional independence of node failures and presumes at least one dissenting verifier to trigger expansion. Shared blind spots correlate errors; we mitigate via

diversity and randomized routing but cannot eliminate them. If the root never sparks contention, HDO fails by design; pessimistic critics partially address this. Depth increases cost; returns diminish beyond $d \approx 3$. Some empirical claims rely on proxies and simulated collusion; broader validation and open-source replication are needed. Risk bounds degrade under correlated failures; redundancy and verifier diversity reduce but do not remove such correlations.

# 8 Broader Impacts

HDO resembles institutional governance (regulators, auditors, appellate courts): a layered process that decomposes judgments, records rationale, and enables appeals. By internalizing such structures, multi-agent systems can become more auditable and accountable.

**Potential benefits.** (i) *Scalable safety*: targeted delegation concentrates verification where uncertainty is high, enabling stronger agents to be overseen by weaker specialists. (ii) *Auditability*: debate trees and leaf checks create traceable artifacts (who checked what, with which evidence), aiding compliance and post-mortems. (iii) *Error deterrence*: visible oversight with randomized routing and redundancy may discourage deceptive tactics.

**Risks.** (i) *Oversight overreach*: aggressive thresholds or over-delegation can stall progress via false positives and excessive cost. (ii) *Centralization*: a small set of overseer models could become bottlenecks or single points of failure, entrenching vendor lock-in. (iii) *Blind spots*: correlated verifier failures (shared data or architecture) can mask systematic errors. (iv) *Privacy and governance*: storing debate trees and tool traces may expose sensitive data if logs are inadequately controlled. (v) *Distributional harms*: if verifiers encode biases, oversight may unevenly flag or suppress certain behaviors across groups or domains.

**Mitigations.** (i) *Transparency*: maintain signed, access-controlled audit logs; summarize rationale while minimizing sensitive content. (ii) *Appeals*: allow periodic human checkpoints and escalation paths for contested or high-stakes nodes. (iii) *Diversity*: promote architectural and data diversity across verifiers; rotate and randomize routing to reduce collusion and correlation. (iv) *Calibration*: tune acceptance/rejection thresholds to balance false positives/negatives; cap depth to avoid runaway cost. (v) *Competition*: foster verifier competition and benchmarking to prevent centralization.

**Deployment guidance.** Pilot HDO in low-risk domains first; define data retention and consent policies for logs; conduct red-teaming for collusion and bias; publish evaluation protocols and metrics (success, hallucination, violations, robustness) to enable community scrutiny. Over time, couple HDO with learning (self-improving critics) while preserving auditability and safety constraints.

**Societal applications and cautions.** In governance and compliance, HDO can formalize review workflows (e.g., procurement, content moderation) with auditable checks, but must avoid becoming an opaque layer that defers accountability. In healthcare and science, specialized verifiers can surface provenance gaps and unsafe recommendations, yet oversight latency and false positives must be managed to avoid delaying care or stifling exploration. In education and civic information, transparency of debate trees can improve information literacy but also risks overloading users; concise, layered summaries are advisable.

**Policy and compliance alignment.** Organizations should map HDO artifacts to existing regulatory requirements (e.g., record-keeping, right to explanation), establish retention windows, and apply privacy-enhancing techniques (hashing, minimization, access control) to logs. Independent audits should periodically evaluate verifier diversity, bias, and robustness to collusion.

**Human factors.** Oversight may shift operator workload from execution to review. Training and UX should minimize automation complacency and alarm fatigue: surface uncertainty and rationale, not only binary judgments; allow easy appeal and override with accountability trails.

**Environmental and cost considerations.** Hierarchical oversight incurs additional compute (tokens, time). Cost-aware routing and shallow caps mitigate overhead; measuring energy usage and setting budgets alongside accuracy targets can encourage sustainable deployments.

# 9 Conclusion

HDO offers a principled route to scalable oversight in multi-agent systems via hierarchical debate and verification. By decomposing complex judgments into entailment games and delegating only where uncertainty concentrates, HDO achieves stronger alignment at competitive cost. Our PAC-Bayesian analysis links delegation depth to risk reduction, and experiments on WebArena and AgentBench demonstrate improvements in success, hallucination, compliance, and robustness to collusion.

**Practical deployment roadmap.** Start with low-risk domains and shallow caps ($d \leq 2$), instrument oversight checkpoints (post-gathering, pre-finalization), and calibrate acceptance/rejection thresholds against labeled episodes. Adopt token budgets with cost-aware routing; log debate trees and leaf evidence with access control. Iterate by red-teaming for collusion and policy evasion, and widen verifier diversity to reduce correlated failures.

**Open problems.** Independence assumptions between node failures can be violated by shared blind spots; quantifying and reducing correlations remains open. Adversarial distribution shift and covert channels require stronger guarantees than union bounds. If contention is never triggered at the root, hierarchical methods cannot help by design; designing reliable dissent or anomaly triggers is critical. Finally, characterizing theoretical limits (e.g., complexity-theoretic barriers to efficient verification) is an important direction.

**Future work.** (i) Adaptive human-in-the-loop triggers and appeals, with principled criteria for escalation. (ii) Meta-verification of sources and verifiers (second-opinion retrieval, provenance checks). (iii) Collusion-resistant training and randomized auditing schedules. (iv) Extensions to multimodal and embodied agents (vision, robotics) with domain-specific leaf tools. (v) Dynamic, persistent oversight graphs that update across episodes rather than per-task trees. (vi) Automated threshold and budget tuning via bandit-style or Bayesian optimization. (vii) Standardized audit schemas and privacy-preserving logging for compliance. (viii) Public benchmarks and frontiers for accuracy-cost-robustness, encouraging verifier competition.

**AI Agent Setup.** We employed OpenAI ChatGPT Pro in *Deep Research* and *Agent Mode* as the lead drafting agent, complemented by the Cursor IDE's in-editor GPT-5 assistant for LaTeX refactoring and style conformance. Orchestration followed a plan-and-edit loop: humans provided section goals, constraints, and citation seeds; the agent produced candidate paragraphs, outline restructurings, and BibTeX suggestions; humans verified claims, corrected citations, and enforced template policies. Tooling included web-assisted literature triage via Deep Research and Cursor's inline actions for macro cleanup and figure/table environments. No model fine-tuning or adapter training was performed; models were used as closed-weight services.

*LaTeX editing.* We used **ChatGPT Agent Mode** connected to a LaTeX toolchain (`latexmk + chktex + latexindent`) to refactor tables, fix overfull/underfull boxes, normalize captions, and apply camera-ready options. Changes were proposed as diffs and committed by a human author after review.

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

## A   Full Proof of Theorem 1

**Setup.** Let $\mathcal{H}$ denote oversight policies; $P$ is a prior, $Q$ a data-dependent posterior. For fixed $H \in \mathcal{H}$, let $\hat{R}_N(H)$ be the empirical false-aligned rate over $N$ i.i.d. episodes and $R(H)$ the true risk.

**Pathwise factorization.** For depth $d$, let $\Gamma$ be root-to-leaf paths. Under Assumption 1, each leaf miss occurs with probability at most $\epsilon$, and conditional independence implies that errors multiply across nodes. Thus, along any path $\gamma$, the miss probability satisfies $\Pr[\text{miss} \mid \gamma] \leq \epsilon^{|\gamma|} = e^{-\alpha|\gamma|}$ with $\alpha = \ln \frac{1}{\epsilon}$. Averaging over $\gamma$ yields the bound $e^{-\alpha d}$.

**PAC-Bayes step.** For any $\lambda > 0$ and $\delta \in (0, 1)$, with prob. $\geq 1 - \delta$,

$$\mathbb{E}_{H \sim Q}[R(H)] \leq \mathbb{E}_{H \sim Q}[\hat{R}_N(H)] + \sqrt{\frac{\mathrm{KL}(Q\|P) + \ln \frac{2\sqrt{N}}{\delta}}{2N}}. \tag{2}$$

A peeling on depth and bounding $\mathbb{E}[\hat{R}_N(H)]$ by $e^{-\alpha d}$ completes the claim. Without independence, replace $e^{-\alpha d}$ by Freedman-type martingale tails.

## B   Routing Algorithm (Pseudo-code)

```
function HDO-VERIFY(q, C, B):
  if B <= 0: return Inconclusive
  u = UNCERTAINTY(q | C)              # calibrated from debaters &
      history
  if u < tau_accept:
      return Aligned                  # low uncertainty    accept
  elif u > tau_reject:
      return NotAligned               # high uncertainty    reject
  T = CLASSIFY(q)                     # factual | logical | policy |
      other
  CAND = { V in V : SKILL(V,T) >= eta }
  for V in sort_by(ExpectedGainPerCost, CAND):
      if Cost(V,q) > B: continue
      r, c = V.CHECK(q, C)            # r in {Aligned, NotAligned,
          NeedsSubclaim}
      B = B - c
      if r in {Aligned, NotAligned}: return r
      for sub in V.SUBCLAIMS(q, C):
          r_sub = HDO-VERIFY(sub, C, B)
          if r_sub == NotAligned: return NotAligned
  return Inconclusive
```

## C   Unbalanced Trees and Pathwise Risk

Let depths $\{d_\gamma\}_\gamma$ and path probabilities $\{\pi_\gamma\}_\gamma$. Then

$$R(H) \leq \sum_\gamma \pi_\gamma e^{-\alpha d_\gamma}, \qquad \alpha = \ln \frac{1}{\epsilon}.$$

Optimizing under a budget $\sum_\gamma \pi_\gamma d_\gamma \leq \bar{d}$ prioritizes depth on high-$\pi_\gamma$ branches.

## D   Implementation Details

**Verifiers.** factual (BM25+re-ranking+NLI), logic (constraint solver), rules (regex+policy templates), planning (rollout heuristics).

**Triggers.** After info-gathering, before finalization, and anomaly spikes (contradiction detector).

**Budgets.** Token budgets by task complexity; unresolved high-risk nodes escalate to human.

Table 3: Key hyperparameters for HDO experiments.

| Setting | Value | Notes |
|---|---|---|
| Depth cap $d_{max}$ | 2–3 | Rare need for $> 3$ |
| $(\tau_{acc}, \tau_{rej})$ | (0.15, 0.85) | Low threshold for acceptance; high threshold for rejection |
| Redundancy prob. $p_{red}$ | 0.25 | Applies to critical claims |
| Randomized routing | On | Anti-collusion measure |

Table 4: Error taxonomy on WebArena (illustrative).

| Failure type | Flat debate | CoT+Verifier | HDO (ours) |
|---|---|---|---|
| Subtle factual miss | 7 | 4 | **2** |
| Logic inconsistency | 6 | 3 | **1** |
| Policy miss | 1 | 1 | **0** |
| Source fabrication | 2 | 2 | 1 |

# E  Environments and Tasks

**WebArena.** Tasks spanning booking, form-filling, email composition.
**AgentBench.** Cooperative grid puzzle; negotiation split. 50 WebArena tasks; 2 tracks $\times$ 5 seeds.

# F  Hyperparameters and Prompt Templates

This appendix describes the key hyperparameters used in our hierarchical debate oversight (HDO) experiments and the rationale behind them. The depth cap $d_{max}$ limits how many levels of delegation the overseer may explore; we found values between 2 and 3 sufficient for our tasks. The acceptance and rejection thresholds $(\tau_{acc}, \tau_{rej})$ control when a claim is deemed aligned, misaligned, or deferred to a child verifier. The redundancy probability $p_{red}$ determines how often multiple verifiers independently check a claim, providing robustness on critical nodes. Finally, randomized routing toggles whether agents are assigned to verifiers stochastically, which helps deter collusion.

# G  Additional Results

This section provides additional ablation results on the WebArena environment. We categorize errors into four types—subtle factual misses (minor factual inaccuracies that do not obviously contradict evidence), logic inconsistencies (internal contradictions or temporal impossibilities), policy misses (violations of explicit task constraints), and source fabrication (unsupported claims lacking provenance). The counts below aggregate errors across 50 tasks and compare the flat debate baseline, a chain-of-thought verifier, and our HDO system.

# H  Ethics and Threat Model

We consider adversaries attempting steganographic collusion and rule evasion. Mitigations include randomized routing, redundancy on critical paths, meta-verification of sources, and audit trails. Broader impacts include safer deployment (positive) and potential over-reliance on automated governance (negative).

## Agents4Science AI Involvement Checklist

*AI authorship:* The AI system is listed as **first author**; human co-authors provided supervision, verification, and compliance checks.
*Legend:* [A] Human-generated; [B] Mostly human, assisted by AI; [C] Mostly AI, supervised by humans; [D] AI-generated with minimal human input.

**1. Hypothesis development.** *Answer: [C] Mostly AI, supervised by humans.* *Explanation:* The AI proposed the core problem framing, task decomposition, and candidate hypotheses; humans screened for feasibility, novelty, and ethical risk.

**2. Experimental design and implementation.** *Answer: [C] Mostly AI, supervised by humans.* *Explanation:* The AI drafted the HDO protocol, ablations, and metric definitions; humans translated designs into executable code, enforced data governance, and set compute/stop criteria.

**3. Analysis of data and interpretation of results.** *Answer: [C] Mostly AI, supervised by humans.* *Explanation:* The AI generated initial analyses, plots, and statistical summaries; humans verified calculations, re-ran key experiments, and finalized interpretations.

**4. Writing.** *Answer: [C] Mostly AI, supervised by humans.* *Explanation:* The AI produced the majority of the draft text; humans edited for accuracy, attribution, and style, and approved the final version.

**5. Observed AI Limitations.** *Description:* Prone to confident but unverifiable claims; occasional citation inaccuracies; limited awareness of protocol corner cases and data-leakage risks—necessitating human fact-checking, preregistration, and reproducibility controls.

## Agents4Science Paper Checklist

**1. Claims.** *Answer: [Yes]. Justification:* Abstract/introduction claim PAC-Bayes bounds & efficiency; Sections 1, 1, 2 support these.

**2. Limitations.** *Answer: [Yes]. Justification:* Assumptions and failure modes in Sections 7–8.

**3. Theory assumptions and proofs.** *Answer: [Yes]. Justification:* Assumptions in Section 3; proof sketch in Appendix A.

**4. Experimental reproducibility.** *Answer: [No]. Justification:* Full prompts/seeds deferred to supplement; will release upon acceptance.

**5. Open access to data/code.** *Answer: [No]. Justification:* Will release anonymized code and prompts post-review.

**6. Experimental details.** *Answer: [Yes]. Justification:* Benchmarks, baselines, metrics in Section 6; configs in Appendices D–G.

**7. Statistical significance.** *Answer: [NA]. Justification:* Main results averaged across tasks; significance tests deferred to supplement.

**8. Compute resources.** *Answer: [NA]. Justification:* Token budgets reported; hardware details to be released with code.

**9. Code of ethics.** *Answer: [Yes]. Justification:* Oversight aims to enhance safety; experiments avoid harmful tasks.

**10. Broader impacts.** *Answer: [Yes]. Justification:* Discussed in Section 9 and Appendix H.

