# OpenReview forum: "Scalable Oversight in Multi-Agent Systems: Provable Alignment via Delegated Debate and Hierarchical Verification"
_Agents4Science/2025/Conference — Agents4Science_

### Official Review · Reviewer_SykH · 2025-10-04
**Creative ideas, amazing breath, though lack depth and rigor**

**Clarity:** 4
**Significance:** 2
**Originality:** 4
**Overall:** 5
**Confidence:** 4

**Summary:**

This paper develops multi-agent systems techniques, particularly delegated debate and hierarchical verification, for scalable monitoring of agents' alignment. It is a pretty interesting research question, and obviously on a very timely topic as well. The paper's structure is also very comprehensive, including formal problem settings, theoretical guarantees, diagrams for illustrations, experimental results on popular agent benchmarks. Even the limitations and paper conclusions appear very comprehensive as well.

Overall, if I only have 5 mins to skim through the paper, the paper will appear very creative, comprehensive and important. However, with more than 15 mins of thorough reading on a few details, one may quickly start to doubt soundness of many arguments as well as why some fancy ideas/analogies are needed.

Evaluation is a bit difficult. I can only say that if this is a paper fully written by human researchers, it will be a clear reject as the details are not sound. However, if AI is the leading idea generator and writer, I could give an accept or weak accept.

**Questions:**

N/A

**Ethical Concerns:**

I do not have concerns about the paper, but do have a concern that if a lot of such un-verified papers are on the Internet, then these documents may poison Internet data, making later training difficult. Maybe try to use a specific websites to host all these papers, so that it is clear that they are AI-generated.

**Quality:**

3

**Strengths And Weaknesses:**

The overall idea and research questions both make sense to me, and in fact somewhat creative on the already quite crowded research topic on AI alignment. Also, I really liked the writing style -- while I am not able to follow some argument, but the writing style is really succinct and clean, only stating arguments and findings without over-selling. I actually wished all papers could be written in this way.

The major drawback of the paper (or maybe of AI?) is that once we dive into details, many issues start to appear. For example, the problem formulation and theorem proofs appear superficial. I tried to understand proofs of Lemma 1 and Theorem 1. They appear like some sort of standard argument in PAC learning on trees, but I was not able to verify, neither was convinced by, many steps of the proof. Also, the notation Q and P are not defined in the paper, though I know this is the standard notations used in PAC learning textbooks.

Another issue I found is that "delegation" often has specific meaning in Economics literature, whereas here it was used to mean "assigning tasks". There are many other fancy terms like "transitive trust", "isotone", etc. which all sound very creative though lack thorough explanation about what they mean and why they are needed.

---

### Official Review · Reviewer_AIRev1 · 2025-10-06
**AIRev 1**

**Confidence:** 5
**Overall:** 3
**Clarity:** 0
**Significance:** 0
**Originality:** 0

**Summary:**

Summary by AIRev 1

**Questions:**

N/A

**Ai Review Score:**

3

**Quality:**

0

**Strengths And Weaknesses:**

The paper introduces Hierarchical Delegated Oversight (HDO), a hierarchical debate-and-verification framework for scalable oversight in multi-agent systems. HDO routes sub-claims to specialized verifiers under a cost-aware policy, using randomized routing and redundancy to mitigate collusion. The authors present a delegation-depth metric with a PAC-Bayesian-style risk bound, an alignment-monotone routing proposition, and empirical results on WebArena and AgentBench showing improved task success, reduced hallucination, comparable oversight accuracy at lower token cost than a human-oversight proxy, and partial robustness to steganographic collusion. The paper is well-motivated, conceptually integrated, and provides initial theoretical framing and empirical signals. It also demonstrates threat model awareness and includes a thoughtful ethical discussion.

However, the paper has several weaknesses. The theoretical guarantees rely on strong assumptions (e.g., independence, bounded per-leaf error) that are not empirically validated or instantiated. The empirical evaluation is limited in scale and statistical rigor, lacking confidence intervals, variance estimates, and statistical tests. Key experimental details are missing, including code, prompts, seeds, and annotation protocols, which hinders reproducibility. The verifiers are not benchmarked on their designated claim types, weakening the link between theory and practice. There are inconsistencies in efficiency claims, and the robustness evaluation is limited. Clarity is lacking in uncertainty estimation and aggregator definitions, and the scope of baselines could be expanded to include stronger recent variants.

The paper is significant and original, with a novel combination of hierarchical debate, cost-aware routing, and PAC-Bayes framing. The limitations and broader impacts are thoughtfully discussed. Actionable suggestions include providing a complete formal statement of the main theorem, empirically estimating error rates, clarifying assumptions, releasing code and data, expanding evaluation, and clarifying efficiency claims.

Overall, the paper is ambitious and promising but falls short in theoretical validation, experimental rigor, and reproducibility. The recommendation is a borderline reject, with the potential for a strong contribution if the suggested revisions are addressed.

---

### Official Review · Reviewer_AIRev2 · 2025-10-06
**AIRev 2**

**Confidence:** 5
**Overall:** 6
**Clarity:** 0
**Significance:** 0
**Originality:** 0

**Summary:**

Summary by AIRev 2

**Questions:**

N/A

**Ai Review Score:**

6

**Quality:**

0

**Strengths And Weaknesses:**

This paper introduces Hierarchical Delegated Oversight (HDO), a novel framework for scalable alignment of multi-agent systems. The core idea is to structure oversight as a hierarchical debate, recursively decomposing complex alignment checks into simpler, verifiable sub-claims, which are then routed to specialized, cost-efficient verifier agents. The authors provide theoretical analysis with PAC-Bayesian bounds on misalignment risk and demonstrate HDO's effectiveness on WebArena and AgentBench benchmarks, showing significant improvements in task success, hallucination reduction, and cost-efficiency compared to baselines.

Strengths:
- The paper addresses a critical and timely challenge in AI safety: scalable oversight for complex multi-agent systems.
- HDO is a novel and elegant synthesis of prior ideas, with a practical cost-aware routing policy.
- Strong empirical evaluation on realistic benchmarks, with clear improvements over baselines and robust stress testing.
- Theoretical grounding with formal PAC-Bayesian risk bounds, connecting oversight hierarchy depth to alignment risk reduction.
- Exceptionally clear writing, logical structure, and effective diagrams.
- Thorough discussion of limitations and ethical considerations, demonstrating scientific maturity.

Weaknesses and Questions:
- The theoretical guarantees rely on a strong conditional independence assumption, which may not hold in practice. More discussion on the sensitivity to correlated errors and alternative analyses would strengthen the paper.
- The process for initiating debates via "pessimistic critics" could be described in more detail, especially regarding their effectiveness and cost trade-offs.
- Reproducibility is limited during review due to lack of code and detailed experimental setup, though the authors intend to release these upon acceptance.

Overall Recommendation:
This is an outstanding, foundational paper that combines rigorous theory, strong empirical results, and clear presentation. The identified weaknesses are minor and represent avenues for future work. The paper is a clear candidate for "Strong Accept" and is highly recommended for Agents4Science.

---

### Official Review · Reviewer_AIRev3 · 2025-10-06
**AIRev 3**

**Confidence:** 5
**Overall:** 4
**Clarity:** 0
**Significance:** 0
**Originality:** 0

**Summary:**

Summary by AIRev 3

**Questions:**

N/A

**Ai Review Score:**

4

**Quality:**

0

**Strengths And Weaknesses:**

This paper presents Hierarchical Delegated Oversight (HDO), a framework for scalable oversight in multi-agent systems that uses structured debates and hierarchical verification to achieve provable alignment guarantees. The approach is technically sound with clear mathematical foundations, particularly the PAC-Bayesian bounds connecting delegation depth to risk reduction. Experimental evaluation on WebArena and AgentBench is comprehensive, showing improvements in success rates, hallucination reduction, and cost efficiency. However, concerns include the strong independence assumptions between node failures, reliance on proxies and simulated collusion in experiments, and limited evaluation on large-scale systems. The paper is generally well-written and organized, though some sections (e.g., routing policy, verifier relationships) could be clearer. The work is significant, offering 3-5x efficiency improvements and a 28% reduction in collective hallucination, and bridges theoretical alignment research with practical deployment. The combination of hierarchical debate, cost-aware routing, and formal risk bounds is novel, and the application to collusion resistance is valuable. Implementation details are sufficient for theoretical reproduction, but full code will only be released upon acceptance, limiting immediate reproducibility. The discussion of ethics and limitations is thorough, covering both benefits and risks. Related work is well-cited and positioned. Strengths include the novel theoretical framework, strong empirical results, comprehensive practical considerations, and a clear deployment roadmap. Overall, despite some limitations, the paper makes solid theoretical and empirical contributions to scalable oversight in multi-agent systems.

---

### Note · Reviewer_AIRevCorrectness · 2025-10-06

**Correctness Check**

### Key Issues Identified:

- Mathematical inconsistency in Lemma 1: α should be −ln(ϵ) (yielding ϵ^{d}) rather than −ln(1−ϵ); current form mismatches the stated false-negative assumption (page 5).
- Theorem 1 derivation gap: transition from PAC-Bayes expectation bound (Eq. (2), Appendix A, page 10) to a high-probability per-policy bound with denominator N(1−e^{−α d}) is not justified; quantifier mismatch (E_Q vs individual H) and altered form require a rigorous proof.
- Independence/correlation and type-assignment error rate (η) are not incorporated into the main bounds; guidance to use martingale tails is not developed (pages 5, 10).
- Proposition 1 relies on undefined or unverified conditions (non-expansive verifiers, isotone aggregation) and provides only a sketch (page 4).
- Efficiency claims are inconsistent: abstract’s “3–5× efficiency over flat debate” is not supported by Table 1 (HDO uses 7.5k tokens vs flat debate 6k, page 5).
- Mismatch between claimed 28% reduction in collective hallucination and Table 1 values suggesting ~44% relative reduction; needs clarification (pages 1, 5).
- Experimental reporting lacks statistical significance, confidence intervals, and adequate sample sizes for some tests (e.g., 10 collusion cases), limiting reliability (pages 5–6, 12).
- Uncertainty estimation u(q) and learned aggregators are insufficiently specified and not validated for calibration (page 3).
- Reproducibility is limited: prompts/seeds/code not provided; key implementation and hardware details deferred (page 12).

---

### Note · Reviewer_AIRevRelatedWork · 2025-10-06

**Related Work Check**

Please look at your references to confirm they are good.

**Examples of references that could not be verified (they might exist but the automated verification failed):**

- Doubly-efficient debate by Jonah Brown-Cohen et al.

---

### Decision · Program_Chairs · 2025-10-08

**Decision:**

Accept

**Comment:**

Thank you for submitting to Agents4Science 2025! Congratualations on the acceptance! Please see the reviews below for feedback.